# Speciation on the Roof of the World: Parallel Fast Evolution of Cryptic Mole Vole Species in the Pamir-Alay—Tien Shan Region

**DOI:** 10.3390/life13081751

**Published:** 2023-08-16

**Authors:** Aleksey Bogdanov, Valentina Tambovtseva, Sergey Matveevsky, Irina Bakloushinskaya

**Affiliations:** 1Koltzov Institute of Developmental Biology, Russian Academy of Sciences, 119334 Moscow, Russia; bogdalst@yahoo.com; 2Vavilov Institute of General Genetics, Russian Academy of Sciences, 119991 Moscow, Russia; sergey8585@mail.ru

**Keywords:** speciation, cryptic species, Robertsonian translocations, mitochondrial and nuclear gene variability, *Ellobius*, Mammalia, Rodentia

## Abstract

Speciation is not always accompanied by morphological changes; numerous cryptic closely related species were revealed using genetic methods. In natural populations of *Ellobius tancrei* (2*n* = 54–30) and *E. alaicus* (2*n* = 52–48) of the Pamir-Alay and Tien Shan, the chromosomal variability due to Robertsonian translocations has been revealed. Here, by comprehensive genetic analysis (karyological analyses as well as sequencing of mitochondrial genes, *cytb* and *COI*, and nuclear genes, *XIST* and *IRBP*) of *E. alaicus* and *E. tancrei* samples from the Inner Tien Shan, the Alay Valley, and the Pamir-Alay, we demonstrated fast and independent diversification of these species. We described an incompletely consistent polymorphism of the mitochondrial and nuclear markers, which arose presumably because of habitat fragmentation in the highlands, rapid karyotype changes, and hybridization of different intraspecific varieties and species. The most intriguing results are a low level of genetic distances calculated from mitochondrial and nuclear genes between some phylogenetic lines of *E. tancrei* and *E. alaicus,* as well significant species-specific chromosome variability in both species. The chromosomal rearrangements are what most clearly define species specificity and provide further diversification. The “mosaicism” and inconsistency in polymorphism patterns are evidence of rapid speciation in these mammals.

## 1. Introduction

New species emergence is not a one-step process, even in cases of so-called sudden speciation. Historically, the origin of a new species has been considered as a gradual process; for a long time, it was thought that only one way was possible—geographically separated populations gradually accumulate changes to some critical level, after which taxonomists (with some degree of subjectivity) are able to describe them as distinct species [1,2]. The concept that divergence increases linearly (i.e., at a constant rate over time) has practically become a dogma; indeed, the molecular clock model is also based on this [3,4]. Linearity is ensured by the interaction of selection and drift. Adopting this approach does not allow us to hope to observe the appearance of new species in nature. However, models based on the study of closely related species extend our understanding of the patterns of speciation [5]. Cases of “young” species with similar ecological and phenotypic peculiarities, compounded by cases of hybridization due to secondary contact, are of particular interest.

In the middle of the 20th century, fossorial cricetid rodents of the *Ellobius* genus were thought to be a rather old group with only one or two species [6]. Now the genus consists of five species in two subgenera: *Ellobius* and *Bramus*. The subgenus *Bramus* is represented by two species: the Transcaucasian mole vole, *E. lutescens* Thomas, 1897, and the southern mole vole, *E. fuscocapillus* Blyth, 1843. The subgenus *Ellobius* includes the northern mole vole, *E. talpinus* Pallas, 1770, the eastern mole vole, *E. tancrei* Blasius, 1884, and the Alay mole vole, *E. alaicus* Vorontsov et al., 1969 [7,8]. Meanwhile, the subgenus *Ellobius* currently includes three species, although it has previously been suggested that the forms from Mongolia can be considered as independent species [8]. The northern mole vole *E. talpinus* is characterized by high mtDNA variability and lack of chromosomal polymorphism [9]. The majority of phylogenetic lines of *E. tancrei* are chromosomally stable (2*n* = 54), while the populations from the Surkhob Valley carry numerous Robertsonian translocations, which have recently been described by Zoo-FISH in three low-chromosomal forms and numerous hybrids with the original 54-chromosome form [10]. The Alay mole vole *E. alaicus* obtained less Rbs (Robertsonian translocations) and, as we demonstrated previously, translocations originated independently in these closely related species [11,12]. With no known fossil record for mole voles in Tien Shan and Pamir-Alay, a very approximate divergence time for *E. tancrei* and *E. alaicus* has been estimated as less than 180 kya by genetic data [8]. Chromosomal variability, as well as the specificity of different DNA markers in these species, suggests extremely rapid diversification. Notably, within approximately 30 years, i.e., one generation of researchers, it was possible to record the change and fixation of the karyotype in *E. alaicus* population in Tajikistan [11]. Such rate of genetic changes can be explained primarily by the subterranean lifestyle, sociality, and higher inbreeding tolerance [13], which maintains a low effective population size in these species.

In this paper, we analyze extensive material from various parts of the range of the two cryptic “young” species, including areas of secondary contacts in distinct mountain systems. Closely related species of mole voles inhabit the Pamir-Alay, Alay Valley, and Tien Shan, which are part of the high mountain systems of Central Asia, metaphorically called the “roof of the world”. The objective of this study was to demonstrate that these two closely related species of mole voles, *E. tancrei* and *E. alaicus*, evolve independently of each other, in turn forming different phylogenetic lines due to rapid changes in chromosome sets and geographical fragmentation.

## 2. Materials and Methods

### 2.1. Samples

In total, 26 mole voles were collected in 10 localities in the Tien Shan and the Alay Valley (southeastern Kyrgyzstan) in 2022. Animals were treated according to conventional international protocols according to the Guidelines for Humane Endpoints for Animals Used in Biomedical Research. All the experimental protocols were approved by the Ethics Committee for Animal Research of the Koltzov Institute of Developmental Biology RAS in accordance with the Regulations for Laboratory Practice in the Russian Federation, the most recent protocol being № 37-25.06.2020. Every possible care was taken to reduce the animal’s suffering during capturing and sampling. Preliminary species identification was performed by karyotyping. We also used previously studied specimens of *E. alaicus* and *E. tancrei* from Kyrgyzstan and Tajikistan [12] for the species identification and determination of detailed pattern of inter- and intraspecific genetic variability. All data samplings are presented in Appendix A and the capture points of the mole voles are shown in Figure 1. The total number of used samples was 82. Tissue samples and chromosome suspensions were obtained in our expeditions in 2008, 2010, 2013, and 2015–2022 field seasons for the Large-Scale Research Facility “Collection of wildlife tissues for genetic research” IDB RAS, state registration number 3579666.

### 2.2. Karyotyping 

For almost all newly collected animals (25 specimens), we made chromosome analysis (Appendix A). Chromosomes were prepared from bone marrow according to [14]. G-banding was achieved using trypsin digestion [15]. For identifying the chromosomes, especially the Robertsonian translocations, we used karyograms previously proved by Zoo-FISH [11]. Routine stained and G-banded metaphase plates were captured with a CMOS camera, mounted on an Axioskop 40 microscope (Zeiss, Oberkochen, Germany). Images were processed in Paint Shop Pro X2 (Corel).

### 2.3. Sequencing

Total DNA from all 26 new samples was isolated by phenol-chloroform deproteinization after treatment of shredded tissues (heart, muscle, or kidney) with proteinase K [16]. As in our previous publication [12], we analyzed the complete mitochondrial cytochrome *b* gene (*cytb*), two overlapping fragments of the nuclear interphotoreceptor retinoid-binding protein gene, exon 1 (*IRBP*), and two non-overlapping fragments of the nuclear X-inactive specific transcript gene (*XIST*); primers and conditions for polymerase chain reaction (PCR) were the same as those described in the [12]. Moreover, in the total sample (in all 82 animals), we sequenced most of the mitochondrial cytochrome *c* oxidase subunit 1 gene (*COI*) for the first time. For the *COI* fragment amplification, we used the specially designed primers: COI-AF (5′-CCT-CTG-TGC-TTA-GAT-TTA-CAG-TC-3′) and R1182-Etal (5′-CCT-GTG-AAT-AGT-GGG-AAT-CAG-TG-3′). PCR was conducted in ABI Veriti 96 thermal cycler (Applied Biosystems, Waltham, MA, USA) in a mixture containing 25 ng DNA, 3 μL 10×Taq Red buffer, 0.3 μL 10 mM dNTP solution, 4 pM of each primer, 0.15 μL of Taq polymerase (5 units/μL), and deionized water to a final volume of 9.65 μL. Amplification was as follows: preheating at 95 °C for 4 min, and then 5 cycles in a sequential mode of 40 s at 95 °C, 1 min at 63 °C, and 1 min at 72 °C; this was followed by 34 cycles in a sequential mode of 40 s at 95 °C, 1 min at 62 °C, and 1 min at 72 °C. The reaction was completed by a single final elongation of PCR products at 72 °C for 4 min. Automatic sequencing was carried out using ABI PRISM^®^ BigDye^TM^ Terminator v. 3.1 Kit with AB 3500 genetic analyzer (Applied Biosystems, Waltham, MA, USA) at the Core Centrum of the Koltzov Institute of Developmental Biology RAS.

Accession numbers of sequences deposited to the GenBank were as follows: OR231548–OR231573 for the *cytb* gene, OR232598–OR232679 for the *COI* gene, OR231574–OR231599 for the *IRBP* gene, OR231600–OR231625 for the *XIST* gene first fragment, and OR231626–OR231651 for the *XIST* gene second fragment. We took into account previously published sequences [8,11,12], which were included in this study (MG264322, MG264324, MG264326–MG264330, MG264345, MG264346, MK544901, MK544903, MK544904, MK544906–MK544917, MT468380, ON333901–ON333931, MT478768–MT478770, ON333848–ON333900, MK544921–MK544926, ON314941–ON314990, and ON314885–ON314940, see the Appendix A). Preliminary species identification was performed on the basis of the karyotyping results. A total of 82 specimens were used for the statistical analyses of variability of the *cytb*, *COI*, *IRBP*, and *XIST* genes as well for building haplotype networks and phylogenetic trees.

### 2.4. Molecular Evolutionary Analyses

The alignments of DNA sequences were made with the MUSCLE algorithm [17] in MEGA X software [18]. The lengths of studied sequences were 1143 bp for the *cytb* gene, 1077 bp for the *COI* gene fragment, the *IRBP* gene fragment was 1404 bp, and the two non-overlapping parts of the *XIST* gene consisted of 444–448 and 525–526 bp (969–973 bp in total). Various lengths of the *XIST* gene fragments were due to deletions in sequences of several specimens. Before statistical analysis, the non-extended deletions were filled in by random nucleotides, which were absent in the total mole vole sample; this solution was applied in our previous publication [12]. The aligned fragments of the *XIST* gene were joined and analyzed as one sequence. The sequences of the protein-coding gene fragments (*COI* and *IRBP*) were presented by entire codons, i.e., started from first nucleotide and finished by third nucleotide of some codons. Sites, in which two overlapped peaks reproducibly observed on the chromatograms of the nuclear genes, were coded and treated as heterozygous. Single heterozygous sites were ignored in statistical treatment. If nuclear sequence sample was represented by two distinct homozygous genotypes, differing in several sites, as well genotypes, which were heterozygous in all these sites simultaneously, separation of the heterozygous (“hybrid”) genotypes into haplotypes was carried out “by hand” as it was described earlier [12]. Otherwise, the phasing of polymorphic loci in heterozygous genotypes was carried out based on combinations of alleles in homozygotes. For each case, we analyzed known haplotypes of the surrounding populations to exclude random alternation of heterozygous sites [19].

The uncorrected mean and pairwise genetic *p*-distances (D) between mole vole species and intraspecific groups as well within them were calculated using Mega X software (https://www.megasoftware.net/) [18]. For all molecular markers, trees were built based on the Maximum Likelihood (ML) method using IQ-Tree software, version 2.0-rc2 [20,21]; the ModelFinder option [22] was applied to get optimal model evaluation of nucleotide substitutions for each gene. Along with study of each gene polymorphism separately, we conducted an analysis of the concatenated *cytb* and *COI* sequences, partitioning the dataset by gene and applying the gene-specific substitution models selected using ModelFinder [23]; each partition had its own set of branch lengths. Standard nonparametric bootstrapping was conducted throughout 1000 pseudo replications for all reconstructions.

Furthermore, taking into account a quite high variability of both studied mitochondrial genes and closeness of compared species, Bayesian inference for *cytb* and *COI* genes as well their combined sequence was additionally evaluated in MrBayes software, version 3.2.7 [24,25] to assess the similarity of ML and Bayesian tree topologies. Bayesian analyses were run for 1 million generations with Markov chains sampled every 1000 generations, and 25% of trees were discarded (“burn-in”). Node support was calculated using posterior probability values. The analysis included two independent runs. The Tracer 1.7.1 software [26] was applied to check for convergence and determine the necessary burn-in fraction, which was 10% of the chain length. In all calculations, the effective sample size exceeded 200 for all estimated parameters.

The images of phylogenetic trees were rendered in FigTree 1.4.4 (http://tree.bio.ed.ac.uk/software/figtree/ (accessed on 25 November 2018)) and processed in Paint Shop Pro X2 (Corel). All trees were rooted on the *E. tancrei* samples from Mongolia as the most distant comparing with other populations.

Haplotype networks (haplowebs) were made using the HaplowebMaker (https://eeg-ebe.github.io/HaplowebMaker/ (accessed on 11 July 2020)) [27]. The networks were constructed using the median joining algorithm. The area of the circle corresponded to the number of individuals carrying a particular haplotype. The curved lines denoted hybrid individuals carrying two different haplotypes; the thickness of the curves was proportional to the number of hybrid individuals. Mutations are indicated by dashes on the straight lines connecting the haplotypes.

## 3. Results

### 3.1. Karyological Study 

Among mole voles from the Inner Tien Shan, chromosome analysis demonstrated a presence of animals with two karyotypes predominately. All specimens from the northern shore of the Song Kel Lake (#30, 31), and localities to the east, in the northern Naryn district (#32, 33) had 2*n* = 54 (Figure 1, Appendix A), as *E. tancrei* over most of the species’ range. Locality #32 was a repeat of the same location, where we previously described mole voles with 2*n* = 53, 1 Rb (1.3), which were treated as possible hybrids of *E. tancrei* and *E. alaicus* [11]. In the Fergana Ridge (to the south of Kazarman, the left bank of the Naryn River, #1) and two points of the Naryn district (Bosogo, #2, and At-Bashy, #4), we revealed mole voles with typical for *E. alaicus* 2*n* = 52, 2 Rb (2.11) (Figure 1 and Appendix A, Appendix A). However, one of the specimens collected in #4 had 51 chromosomes in its karyotype with 2 Rb (2.11), 1 Rb (1.3) (Figure 1 and Appendix A, Appendix A). Furthermore, in Naryn district, on the left bank of the Small Naryn River (#3), we found a mole vole with 2*n* = 48, 2 Rb (2.11), 2 Rb (1.3), 1 Rb (6.8), 1 Rb (5.12) (Figure 1 and Figure 2; Appendix A).

### 3.2. Molecular Study

#### 3.2.1. Variability of Mitochondrial Genes

The analysis of mitochondrial *cytb* gene in the total sample of mole voles (including previously studied ones) demonstrated high differentiation within *E. tancrei* and *E. alaicus*, and those samples from Tien Shan, representing divergent intraspecific genetic forms, made significant contribution to genetic variability of both species. Both ML and Bayesian trees (Appendix A) show distribution of specimens into three major clades. First clade (M), most divergent from the other two (D = 0.041–0.044, see Appendix A), is represented by *E. tancrei* from Mongolia. Second clade consists of *E. tancrei* from Central Asia; the clade is differentiated into two sister subclades: the one (T) includes *E. tancrei* with 2*n* = 30–54 from Tajikistan, the other (TS) joins specimens with 2*n* = 54 from the Inner Tien Shan and the flanked branch, corresponding to an *E. tancrei* specimen from #34, Tashkent. Third clade is divided into two sister subclades too: the first (TS) consists of *E. alaicus* from the Inner Tien Shan, and the other comprises specimens from the Alay Valley and northeastern Tajikistan as well as interspecific *E. alaicus*-*E. tancrei* hybrids from #5 (Gulcha vicinities in the north to the Pamir Highway). It is noteworthy that genetic distances between the subclades in the limits of the second and third clades (0.022–0.023) are comparable with the differences between *E. alaicus* and *E. tancrei* from Central Asia (0.026). 

The trees built on the basis of the *COI* gene fragment analysis (Appendix A) show the same main clades and branches as those, which were identified in the *cytb* gene study, with the exclusion of an “unstable” position of clade, joining *E. alaicus* specimens from Inner Tien Shan, in the Bayesian and ML trees. Polytomy in ML tree built on the *COI* analysis data is discussed below. In the Bayesian reconstruction, this clade is sister in the relation to *E. alaicus* populations from the Alay Valley and Tajikistan that is similar to the *cytb* gene analysis data. However, in the ML tree, it demonstrates polytomy along with *E. alaicus* from the Pamir-Alay and Central Asian *E. tancrei* populations. The analysis of the concatenated sequences of both mitochondrial genes leveled the inconsistency and produced Bayesian and ML trees with similar topologies (Figure 3 and Appendix A), like those in the *cytb* gene study (Appendix A). Despite this, it should be concluded that a clusterization pattern for the intraspecific forms of *E. alaicus* and *E. tancrei* may significantly vary, depending on the different mitochondrial markers.

Likewise for the *cytb* gene, the average genetic distances, which were calculated using the *COI* gene fragment and joined sequence of two mitochondrial markers between *E. alaicus* and *E. tancrei* from Central Asia (0.020 for the *COI* gene and 0.023 for concatenated mitochondrial sequence, see Appendix A), insignificantly excess the differences determined between the major subclades within *E. alaicus* (maximally 0.017 for the *COI* gene and 0.020 for concatenated mitochondrial sequence) and *E. tancrei* from Central Asia (maximally 0.015 and 0.019, respectively).

It is notable that the average genetic distance values calculated within the subclades of *E. alaicus* and *E. tancrei* from Central Asia (Appendix A) show some decline of genetic polymorphism in populations from northeastern Tajikistan in comparison with those from Tien Shan in both species. Based on the parameter, the *COI* gene variability, for example, is the same (D = 0.003) in the *E. alaicus* sample from Tien Shan and in the vast group, joining conspecific populations from northeastern Tajikistan and the Alay Valley as well hybrids from #5. Populations of *E. alaicus* from the Alay Valley have one *COI* gene haplotype only (D = 0.000).

Haplotype networks visualized a picture of the *cytb* and *COI* sequence diversity (Appendix A; for concatenated sequences see Figure 4). Haplotypes of *E. tancrei* were shown in shades of green and cyan while shades of purple-pink and orange-brown marked haplotypes of *E. alaicus*. Blue circles corresponded to individuals from previously described *E. tancrei-E. alaicus* hybrid zone [11,12]. A haploweb based on *cytb* and *COI* sequences made it possible to demonstrate numerous mutational steps between intraspecific forms and species along with the high level of their polymorphism. Like patterns in ML and Bayesian trees, we observed that specimens of *E. tancrei* from Mongolian localities (#35–37, Appendix A) had been separated from Central Asian mole vole populations of both species by more considerable genetic distances than Central Asian populations of *E. tancrei* and *E. alaicus* from each other. The populations of both *E. tancrei* and *E. alaicus* to the north of the Fergana Ridge were separated from the southern groups of conspecific populations by a greater distance than the southern groups of populations from the common node. The level of haplotypic polymorphism is relatively comparable in Central Asian *E. tancrei* and *E. alaicus* populations.

#### 3.2.2. Variability of the Nuclear *XIST* Gene

According to our previous results [12], the analysis of the *XIST* gene made it possible to clearly discriminate Mongolian populations of *E. tancrei* only (#35–37, Appendix A). Other specimens and population groups of *E. alaicus* and *E. tancrei* as well interspecific hybrids from #5 (Gulcha vicinities) formed an unresolved polytomy. However, distinct differentiation in *E. alaicus* into two intraspecific groups (“eastern” and “western”) was obvious on this gene: two haplotypes, differing by a three-nucleotide deletion and several substitutions, were revealed in *E. alaicus* from the Alay Valley in both homozygotic and heterozygotic states. The addition of new material from the Inner Tien Shan did not fundamentally change the overall pattern of ML tree topology (Figure 5). At the same time, *E. tancrei* specimens from this area were characterized by the unique *XIST* haplotype that led to their clusterization to separate group, close to *E. tancrei* from the Pamir-Alay. All *E. alaicus* from the Inner Tien Shan had the *XIST* haplotype with deletion in homozygotic state, like the “eastern form” of this species.

A haplotype network based on *XIST* sequences (Figure 6) demonstrated that representatives of Mongolian populations of *E. tancrei* (#35–37, Appendix A) form the clade, most distant from the common node. Central Asian *E. tancrei* formed a compact clade, not including, however, a specimen from Tashkent, Uzbekistan (#34). In comparison, the intraspecific polymorphism of *E. alaicus* is much higher: the chromosomal form from the Pamir-Alay with 2*n* = 48 (#14–18, Figure 1, Appendix A) is represented by a specific haplotype, as well as individuals from the Inner Tien Shan together with a number of individuals from the Alay Valley. We observed the highest haplotypic polymorphism in the populations from the eastern part of the Alay Valley (#6–10, Figure 1, Appendix A) and from the previously described hybrid zone in Gulcha vicinities, #5 [11,12]. 

The *XIST* gene analysis of these populations revealed nine heterozygous specimens: in the hybrid zone #5, in the Alay Ridge (#6), and in the central part of the Alay Valley (#7, 8, 10) that is shown by the curves on the network. In counterpoint to our previous data [12], the haplotypes of the possible backcrosses from the hybrid zone #5 are significantly skewed toward *E. alaicus*.

#### 3.2.3. Variability of the Nuclear *IRBP* Gene

The gene encoding the Interphotoreceptor Retinoid-Binding Protein (IRBP) was used to resolve mammalian phylogeny at lower levels [8]. As we showed previously [12], the *IRBP* genotypes in all specimens of *E. alaicus* and *E. tancrei* (including individuals from Mongolia) differ by three fixed substitutions. Heterozygosity, if it took place, was observed in all these sites simultaneously that allowed us to divide heterozygotic genotypes into haplotypes. All studied *E. tancrei* and the majority of *E. alaicus* had different *IRBP* haplotypes but in some *E. alaicus* specimens from the Alay Valley, like hybrid mole voles from #5, the haplotype, typical for *E. tancrei*, was revealed in heterozygotic or even homozygotic states. We proposed that the phenomenon might be the result of interspecific hybridization or incomplete haplotype sorting, i.e., ancestral polymorphism manifestation.

As in our prior study, the ML tree built on the results of the *IRBP* gene analysis in the total mole vole sample demonstrates a polytomy with some clusters (Figure 7). The first of them includes Mongolian specimens of *E. tancrei*, while the second one, the most variable, has the majority of *E. tancrei* from Central Asia and several *E. alaicus*. The third cluster is represented by the majority of *E. alaicus* as well as some specimens of *E. tancrei*. 

Despite the addition of new material from the Tien Shan, two main haplotype lines, differing in three sites, are still distinctly identified in *E. alaicus* and *E. tancrei,* which makes it possible to reveal the untypical *IRBP* haplotypes in populations. The haplotype typical for *E. tancrei* was additionally diagnosed in the heterozygotic state in *E. alaicus* from the Fergana Ridge (#1), the eastern point in the Alay Valley (#8), and even from its central part (#9), “terra typica” for *E. alaicus.* It is noteworthy that the “introgression” may happen in the opposite direction: in two *E. tancrei* specimens from Naryn vicinities (#33), we found the *IRBP* haplotype characteristic for *E. alaicus* in the heterozygotic state.

In accordance with the ML tree, the haplotype network demonstrated that the polymorphism of the *IRBP* gene was low (Figure 8). It was represented by two major haplotypes in the studied sample. Only the haplotype of the low-chromosomal (2*n* = 30) population of *E. tancrei* from the Pamir-Alay (#24, Figure 1, Appendix A) was faintly different from the others. The vast majority of animals had one of two haplotypes, but this distribution did not reflect the species identity. Carriers of one of the haplotypes were predominantly *E. alaicus*, but the haplotype was also identified in interspecific hybrids (possible backcrosses) from the hybrid zone (#5) and in some representatives of *E. tancrei* with heterozygous genotypes, where we previously described possible hybrids [11]. Both *E. alaicus* and *E. tancrei* individuals, in both the homo- and heterozygous states, shared another haplotype. Homozygous carriers of this haplotype characteristic of most *E. tancrei* among *E. alaicus* were mole voles with distinct karyotypes inhabited remote localities #8 and #11. Hybrid genotypes among *E. alaicus* were found in specimens from the hybrid zone (#5), the Alay Valley (#8, #10, #12), and the Fergana Ridge (#1) (Figure 1, Appendix A).

## 4. Discussion

Analysis of chromosome (Robertsonian translocations) and gene variability (mitochondrial genes, *cytb* and *COI*, and nuclear genes, *XIST* and *IRBP*) in mole voles *E. alaicus* and *E. tancrei* portrays these species as young and fast-evolving, with similar traits in their radiation. Chromosome changes and habitat fragmentation are the leading factors since the divergence occurs in mountain systems. 

The results of our study indicate that intensive genetic differentiation took place in the populations of mole voles inhabiting the Tien Shan. Within *E. tancrei*, three major lines were distinguished (Figure 3): M—individuals from Mongolian localities (#35–37, Figure 1, Appendix A), T—the main part of the sample from Tajikistan (#19–29), TS—a group of specimens from Tien Shan (#30–33), which, according to mitochondrial variability data, also includes a sample from Tashkent, Uzbekistan as a flanked branch (#34). The species *E. alaicus* is also represented by three main lines: W—a group of populations from the western part of the range (Pamir-Alay), corresponding to the 48-chromosome form (#14–18), E—a group of populations from the Alay Valley (#6–10), TS—a sample from the Inner Tien Shan (#1–4). Populations #11–13 belong to “E” group but demonstrate different features (*IRBP* alleles) so we marked these samples by other colors in the nets. Possible hybrids of *E. tancrei* and *E. alaicus* (#5) were marked as “hybrids?”.

A growing body of research on attempts to distinguish closely related species and forms suggests that different datasets require different rather than standardized approaches [28,29]. Haplotype network analysis, which allows us to work with both single-locus and multilocus models at different sample sizes, is gaining popularity in mammalian phylogenetic studies [30].

Joint haplotype network generated by HaplowebMaker for the concatenated sequences of *cytb* and *COI* genes (Figure 4) shows that the populations of *E. tancrei* and *E. alaicus* from Central Asia appeared to be represented by less compact clusters compared to the previously obtained data [12]. The pattern was altered by the addition of data on mole vole populations of both species from the Inner Tien Shen (#1–4, 30–33). The level of haplotypic polymorphism, which we previously described as higher in Central Asian *E. tancrei* populations and lower in *E. alaicus* populations, appeared to be relatively comparable in light of the new data.

The complex, multilevel character of the presented genetic polymorphism is clearly seen in the multigenic net (see Appendix A and Graphical abstract). This net demonstrates the intermediate position of possible hybrids (#5), lineages, which we can identify as species and intraspecific forms, and some genetic exchanges within the eastern (E) populations of *E. alaicus*. 

The karyotype was a key feature when we studied the Tien Shan (TS) populations of both species because the external characteristics and the nuclear markers appeared to be low specific. Inconsistencies in the topology of trees are present even when using only mitochondrial genes, *cytb,* and *COI* (Appendix A) by which the most distinct differentiation of intraspecific groups of *E. tancrei* and *E. alaicus* was observed. Thus, animals from the hybrid zone in Gulcha (#5), relative to the “W” form of *E. alaicus,* and the Tien Shan populations of this species, relative to its samples from Pamir-Alay, had an unstable position in the ML trees built on the different mitochondrial genes. Haplotype networks constructed by *cytb* and *COI* genes (Figure 4, Appendix A) clearly demonstrate these differences too. In addition, the genetic distances calculated within each species turned out to be comparable with the interspecific differences (Appendix A), without taking into account the significantly distant Mongolian populations of *E. tancrei*, which were proposed to be treated as a different species [8]. 

According to our hypothesis provided in a previous paper [12], *E. alaicus* might arise in a territory to the north from the Alay Ridge. Indeed, comparable or even higher diversity of the mitochondrial haplotypes in the Tien Shan populations of both species in comparison with the Pamir-Alay ones (Appendix A) may indicate the origin of *E. alaicus* and Central Asian *E. tancrei* group in Tien Shan or in a territory adjacent on the south side, from where they further moved to Pamir-Alay by different ways, taking into account an absence of *E. tancrei* in the Kyzylsu–Muksu interfluve now. Difficulties in species diagnostics, which we faced in the case of the Tien Shan mole vole populations, are an indirect support for the proposition. Considering noticeable differences in the mitochondrial genes in the Tien Shan and Pamir-Alay population groups in both *E. tancrei* and *E. alaicus*, the history of these species was probably accompanied by the range gaps, rather than long period of isolation and independent evolution of their populations in these mountain systems. This led to the formation of a divergent but haplotype-poor gene pool for each species in Pamir-Alay; the very low *COI* gene polymorphism in *E. alaicus* sample from the Alay Valley and northeastern Tajikistan reflects it most clearly. Independent mutation accumulation and further cleavage of mitochondrial gene pools in Pamir-Alay populations of *E. alaicus* and *E. tancrei* increased the differences between these species.

Recent studies revealed signatures of positive selection in the *cytb* and *COI* genes associated with a subterranean lifestyle in some Arvicolinae, including mole voles [31,32]. COI Met73Ile, CYTB Thr56Ser, CYTB Ile338Val, and CYTB Ala357Thr were suggested as sites under positive selection. In our samples, just two *E. tancrei* (#21) specimens obtained CYTB Ile338Val mutation, and all studied animals got COI Met73Ile unlike the previously studied individuals of *E. talpinus*, representatives of the third of the closely related species of subgenus *Ellobius* [32]. Thus, our data support the hypothesis of positive selection and selection advantage of this mutation in subterranean animals. The assumption of rapid evolution of this group is further underlined by our data. 

Diversification of *E. tancrei* and *E. alaicus* by karyotypic and studied nuclear markers differs if compared with mitochondrial evolution; the mito-nuclear discordance was described by us earlier [12]. The nuclear *XIST* gene does not allow reliable identification of *E. tancrei* and *E. alaicus*, as well as their hybrids, probably due to the slower evolution of nuclear genes compared to mitochondrial ones [33,34]. However, our previous study of this gene unexpectedly revealed the existence of two distinct intraspecific forms within *E. alaicus* and their hybridization in the Alay Valley [12]. Analysis of new samples of *E. alaicus* from the Tien Shan showed that all of them are homozygous for a peculiar haplotype with a deletion that has not yet been found in any other species of the subgenus *Ellobius* (Figure 5 and Figure 6). The variants of the *XIST* gene are spatially distributed as follows: in the populations of the Tien Shan the haplotype with a deletion was fixed, while in the populations of the northeastern Tajikistan the common haplotype was revealed only. This is very similar to the picture observed for mitochondrial genes, but no traces of hybridization were found by the *cytb* and *COI* analysis among *E. alaicus* individuals in the Alay Valley: in any specimen from there we did not find yet a mitochondrial haplotype, characteristic of the Tien Shan populations. This inconsistency can be explained in two ways. First, we can assume that the form with specific mitochondrial haplotypes characteristic of the Tien Shan populations and the form in which the *XIST* gene haplotype with a deletion was revealed in the homozygous state are not the same, and they originated at different times and in different regions. According to this hypothesis, the appearance of the *XIST* gene haplotype with a deletion could be a relatively late event that occurred in any long-term isolated *E. alaicus* populations in the Alay Valley or near it. Later, when climatic conditions changed to more favorable ones that promoted active dispersal of mole voles, the carriers of the *XIST* haplotype with a deletion spread in different directions. When migrating to the Alay Valley, they formed mixed populations with another intraspecies form having the usual *XIST* gene haplotype, and when moving north, they completely displaced all other haplotypes of this gene from the populations. Since such a scenario seems difficult to implement, another explanation is more likely: differentiation by mitochondrial DNA and the *XIST* gene in *E. alaicus* occurred synchronously (moreover, separation of the Tien Shan and Pamir-Alay population groups by this nuclear marker can also be traced in *E. tancrei*, although less distinctly). Consequently, in this case we must admit that there are limited contacts between Pamir-Alay and Tien Shan intraspecific forms of *E. alaicus* and partial invasion of the Tien Shan form to the Alay Valley, and the absence of its mitochondrial haplotypes in this area is caused by their displacement as a result of hybridization with Pamir-Alay populations. 

The results of the *IRBP* gene fragment analysis clearly demonstrate the existence in *E. tancrei* (including populations from Mongolia) and *E. alaicus* of two groups of haplotypes that can be detected in mole voles both in the homozygous and heterozygous states (Figure 7 and Figure 8). The haplotypes of these groups differ from each other by three fixed substitutions, and the study of additional material from the Tien Shan has not yet found any “intermediate” variants of the *IRBP* gene and has not changed the general pattern of its variability. The haplotypes of each of these two groups are characteristic of the majority of individuals of either *E. alaicus* or *E. tancrei*, but not all. Earlier [12], we proposed two alternative explanations for this gene variability, which could be due to either limited hybridization of *E. alaicus* and *E. tancrei* or incomplete lineage sorting, that is, ancestral polymorphism. The study of the Tien Shan mole vole populations does not allow us to make an unambiguous choice in favor of one or the other hypothesis. Revealing heterozygous genotypes in specimens from the Fergana Ridge (#1) and Naryn district (#33), as well as in specimens from Gulcha vicinities (#5), where previously hybrids were identified by karyotypes (27491, 2*n* = 53 and others, see [11]), may well be the result of interspecific hybridization. However, the totally unexpected finding of the *IRBP* gene haplotype, typical for the majority of *E. tancrei*, in specimens from the Alay Valley—from *terra typica* of *E. alaicus*, #9, and in localities #8, #11, and #12—is more likely explained by the ancestral polymorphism hypothesis. To resolve this dilemma, analysis of additional material from numerous localities in the Alay Valley, surrounding territories, and Tien Shan, as well as the use of other nuclear markers in the analysis, is necessary.

Karyotype 2*n* = 51, 2 Rb (2.11), 1 Rb (1.3) was previously described by us for *E. alaicus* from the southern part of the Naryn district [11]. Translocations Rb (6.8), Rb (5.12) were detected for the first time, and no such Rbs were found in *E. alaicus* or *E. tancrei* (Figure 2). These Rbs indicate chromosome differentiation between *E. alaicus* populations from Tien Shan and Pamir-Alay.

Considering the specific sets of Robertsonian translocations in *E. alaicus* and *E. tancrei*, the karyotype rearrangement may have been the very first and perhaps even the initiating evolutionary event that separated these two species. Rb (2.11) is typical for *E. alaicus*, and this translocation was a main character for the species description [35]. The same translocation was revealed in some *E. tancrei* populations from Tajikistan, but it was never revealed in karyotypes alone, without other Rbs; we have previously suggested the independent origin of this translocation in these species [10,11]. However, further chromosomal rearrangements occurred independently and were recorded in *E. alaicus* and *E. tancrei* locally and in different regions: in *E. alaicus*—in the western part of the Alay Valley (2*n* = 50–51), in the interfluve of Kyzylsu and Muksu (2*n* = 48, 2 Rb (2.11), 2 Rb (4.9), 2 Rb (3.10)), and also independently in the Naryn region (2*n* = 51–48, 2 Rb (2.11), 1–2 Rb (1.3), Rb (6.8), Rb (5.12)), whereas in *E. tancrei* all variations were concentrated in Pamir-Alay (2*n* = 54–30). Except for Rb (2.11), all other translocations, which are characteristic of *E. alaicus,* are not found in *E. tancrei*. Interestingly, the appearance of chromosomal forms with different composition and set of Rbs occurs in both species. With the same 2n, the karyotypes turn out to be different in structure, which may lead to sterility or disorders in possible interspecies hybrids due to partial (monobrachial) homology of Rbs. Translocations of Rb(1.3), Rb(3.10), Rb(4.9), Rb (6.8), and Rb (5.12) are specific for *E. alaicus* and indicate chromosome differentiation with *E. tancrei*.

It is noteworthy that chromosome contacts in meiosis have been described in *E. alaicus* in some localities of the Alay Valley. They are considered as a possible mechanism of Robertsonian translocations [36,37]. However, in none of Tien Shan mole vole populations we were able to find individuals with such chromosome contacts in meiotic prophase I. The discovery of contacts provides evidence of a process of Rbs genesis, which in turn indicates intensive phylogenesis and is consistent with the assumption that *E. alaicus* is a young species with significant differentiation.

Despite the mosaic pattern of karyotypic variability and the isolated pools of chromosomal rearrangements, it should be noted that Central Asian populations of *E. tancrei* and *E. alaicus* apparently share a common predisposition to them, inherited from their ancestral form. It is possible that this hypothetical ancestral form acquired some genetic feature (mutation of one or several genes? changes in telomeres? activation of mobile elements?) determining the potential capacity of the karyotype to Robertsonian translocations. Subsequently, this feature might be realized in distinct population complexes of *E. tancrei* and *E. alaicus* under the influence of some natural physical or biological factors, such as natural radiation or some viral infection. Emergence of Rbs leads to changes of recombination patterns and nuclear architecture and can cause meiotic irregularities in hybrids, which lead to reproductive isolation [38,39,40,41]. Modeling studies determined that purifying selection against changes in synteny manifests itself as selection against unbalanced gametes [42,43]. This purifying selection against chromosomal rearrangements disturbing meiosis can only be overcome by drift in small or divided populations [44,45]. We observe exactly such type of populations in mountain ranges of mole voles. 

Chromosomal and nucleotide variability demonstrates a similar evolutionary pattern in *E. tancrei* and *E. alaicus*, and their radiation is based on an ancestral genetic background, similar to some other animal groups [46,47]. We believe that the initiating event in the divergence of this group was chromosome rearrangements. Such mutations lead to changes in linkage groups, recombination rate, and accelerate genomic evolution [40]. These mole voles species reveal differences in their evolutionary paths, so we can consider them to be distinct, though very young, which is consistent with the newest concept of cryptic species and rapid radiation [48,49,50]. Recently, hybridization and its concomitant phenomena have been increasingly considered not as evidence of incomplete differentiation of forms, but as one of the factors contributing to the unique evolutionary pathway of the groups involved [51,52]. Gene flow, localized or affecting a large part of the range, appears to be possible even between species with significant chromosomal differences [53,54,55]. Nonetheless, the presence of limited gene flow does not have a destructive effect on the integrity of gene pools of *E. tancrei* and *E. alaicus*. Instead, this restricted hybridization emphasizes the distinctiveness of the cryptic species we are studying, which is in line with modern views [56,57] because of geographic disunity and chromosomal changes, as in butterflies [58,59,60]. If we compare the evolutionary patterns of *E. tancrei* and *E. alaicus* with the *Sorex* species group, which is characterized by significant Rb polymorphism too, a particularly rapid rate of evolution in “young” mole vole species becomes apparent, along with the low role of hybridization [61,62].

The combination of multidirectional processes in the history of these species—rapid formation of individual local populations with specific genetic features facilitated by the unique traits of mole vole biology, allopatric formation of new large population groups, and hybridization—makes *E. tancrei* and *E. alaicus* a very intriguing model for studying the early stages of speciation. 

## 5. Conclusions

The study of two cryptic species of the subgenus *Ellobius, E. tancrei* and *E. alaicus*, which inhabit highlands, reveals to us an amazing combination of evolutionary trends: mosaic pattern of chromosomal and molecular genetic variability; asynchronous changes in different markers; exclusive chromosome contacts in meiosis in some populations as the first step to form new Rb translocations; simultaneous representation of zones of hybridization and plausible manifestation of ancestral polymorphism over a vast territory; and parallel fast evolution of two species in separated parts of their ranges. Nevertheless, we are sorely lacking the temporal axis. Paleontological data are virtually unavailable, in addition to inconsistent variability of different molecular markers, making it difficult to calibrate divergence times and to model clade divergence events. Therefore, the question about the “ancestral homeland of mole voles” remains open for the time being. We expect that other types of analysis (e.g., ecological modeling involving ancient climate change data) will allow us to fill this white spot.

## Figures and Tables

**Figure 1 life-13-01751-f001:**
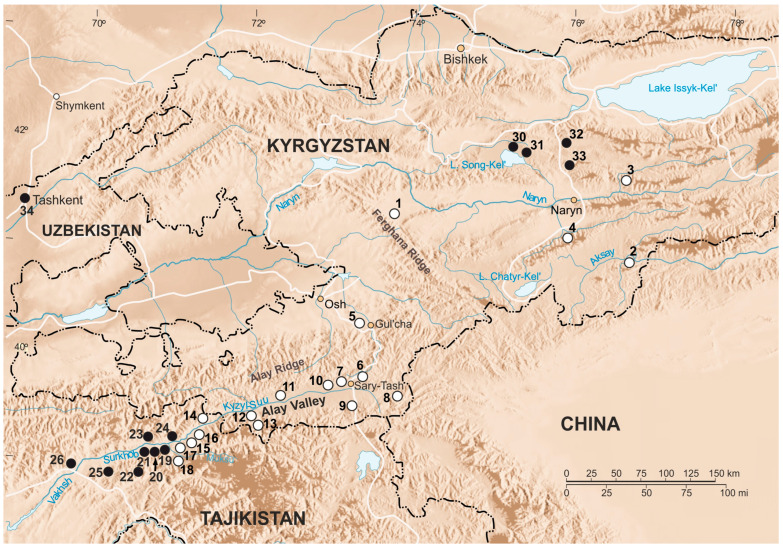
Map of collection points. Collection localities of *E. alaicus* specimens are marked with white circles; collection localities of *E. tancrei* are marked with black circles. The names of the studied localities are given in Appendix A. Points 35–37 are beyond the boundaries of this map fragment.

**Figure 2 life-13-01751-f002:**
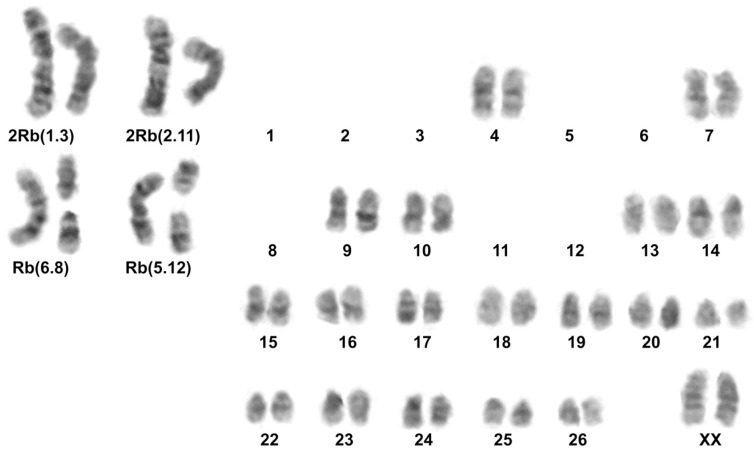
Karyotype of *E. alaicus* 27527, 2*n* = 48, #3; G-banding.

**Figure 3 life-13-01751-f003:**
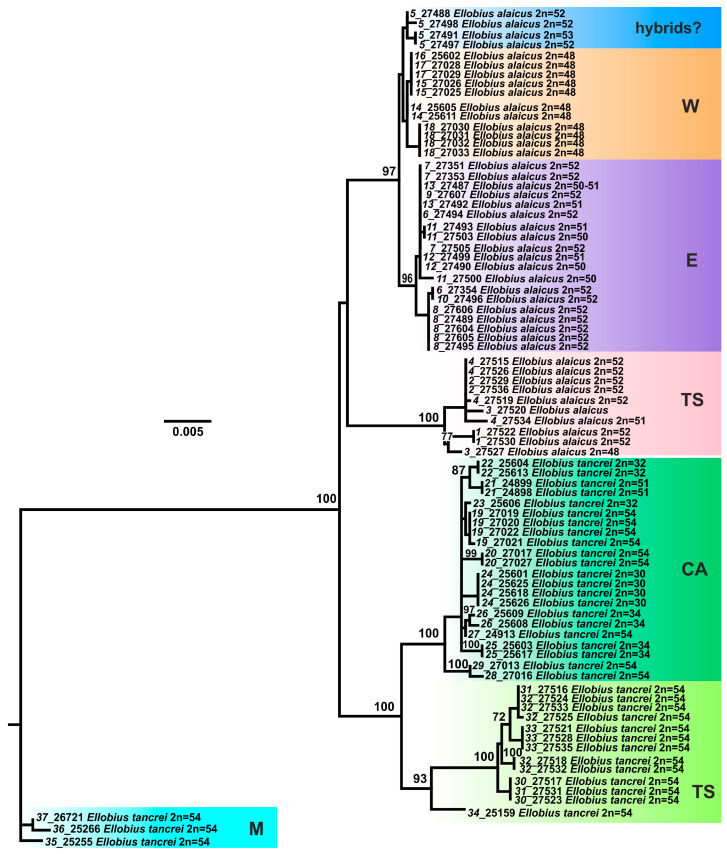
The maximum-likelihood tree based on joined *cytb* gene and *COI* gene sequences of 82 specimens of *E. tancrei* and *E. alaicus*. Samples from *E. tancrei* Mongolian populations were used as an outgroup. Numbers above the nodes correspond to bootstrap support; values <70 are not specified. Sample names are presented as “locality number—individual number”. The color selection corresponds to haplowebs: cyan marks Mongolian (M) *E. tancrei* (#35–37); deep green marks *E. tancrei* specimens from Tajikistan (T) (#19–29); bright green marks Tien Shan (TS) *E. tancrei* (#30–34); blue marks probable interspecific hybrids from (#5); pink marks Tien Shan (TS) *E. alaicus* (#1–4); purple and orange mark “E” (#6–10) and “W” (#14–18) haplotypes of *E. alaicus*, respectively. Populations ##11–13 belong to “E” group but demonstrated some features (*IRBP* alleles).

**Figure 4 life-13-01751-f004:**
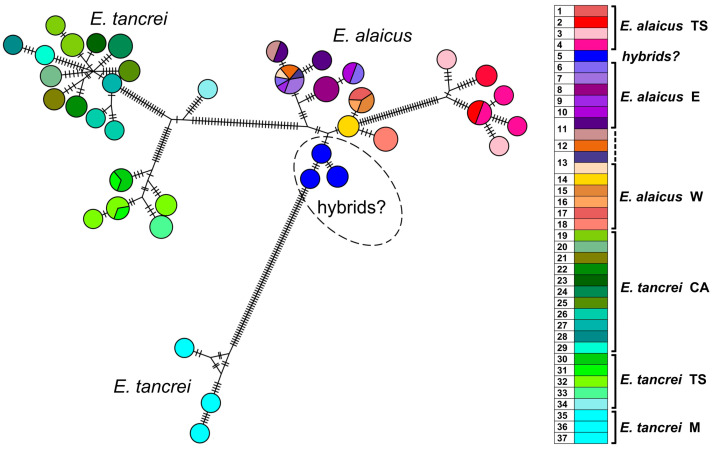
Haplotype network constructed by HaplowebMaker based on the joined sequences of two mitochondrial genes (*cytb* and *COI*). Data from the entire sample are used (82 specimens of *E. tancrei* and *E. alaicus*, see Appendix A). A sample from each locality is colored with its own color; close shades unite species/intraspecific forms according to investigated taxonomic hypotheses. The circles represent haplotypes; the curves between them indicate hybrid individuals carrying both connected haplotypes. The area of the circles and the thickness of the curves are proportional to the number of individuals carrying the haplotype; nucleotide substitutions are indicated by dashes. Dotted line indicates probable interspecific hybrids (#5).

**Figure 5 life-13-01751-f005:**
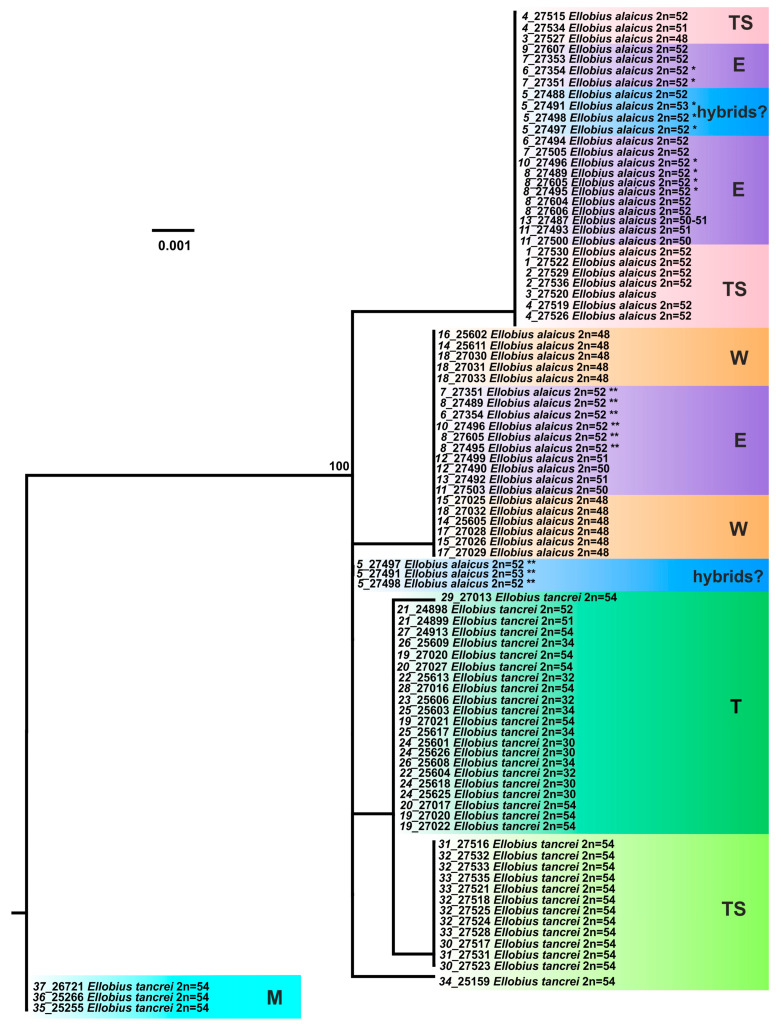
The maximum-likelihood tree based on the *XIST* gene sequences of 82 specimens of *E. tancrei* and *E. alaicus*. Samples from *E. tancrei* Mongolian populations were used as an outgroup. Numbers above the nodes correspond to bootstrap support; values <70 are not specified. Sample names are presented as “locality number—individual number”. The color selection and abbreviations are denoted as specified in Figure 3. Haplotypes that were hypothetically derived from heterozygous genotypes are indicated by asterisks: the haplotype with a deletion is marked by one asterisk, while the other haplotype is marked by two asterisks.

**Figure 6 life-13-01751-f006:**
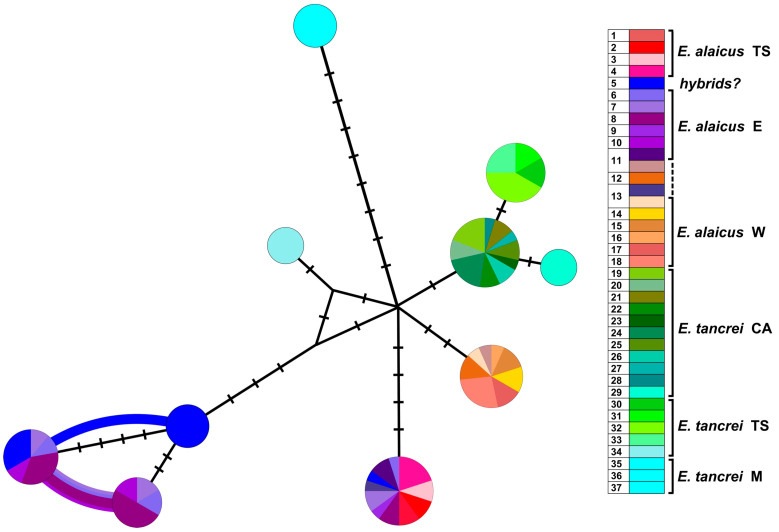
Haplotype network constructed by HaplowebMaker based on the *XIST* gene sequences of 82 specimens of *E. tancrei* and *E. alaicus* (see Appendix A). A sample from each locality is colored with its own color; close shades unite species/intraspecific forms, according to investigated taxonomic hypotheses. The circles represent haplotypes; the curves between them indicate hybrid individuals, carrying both connected haplotypes. The area of the circles and the thickness of the curves are proportional to the number of individuals carrying the haplotype; nucleotide substitutions are indicated by dashes.

**Figure 7 life-13-01751-f007:**
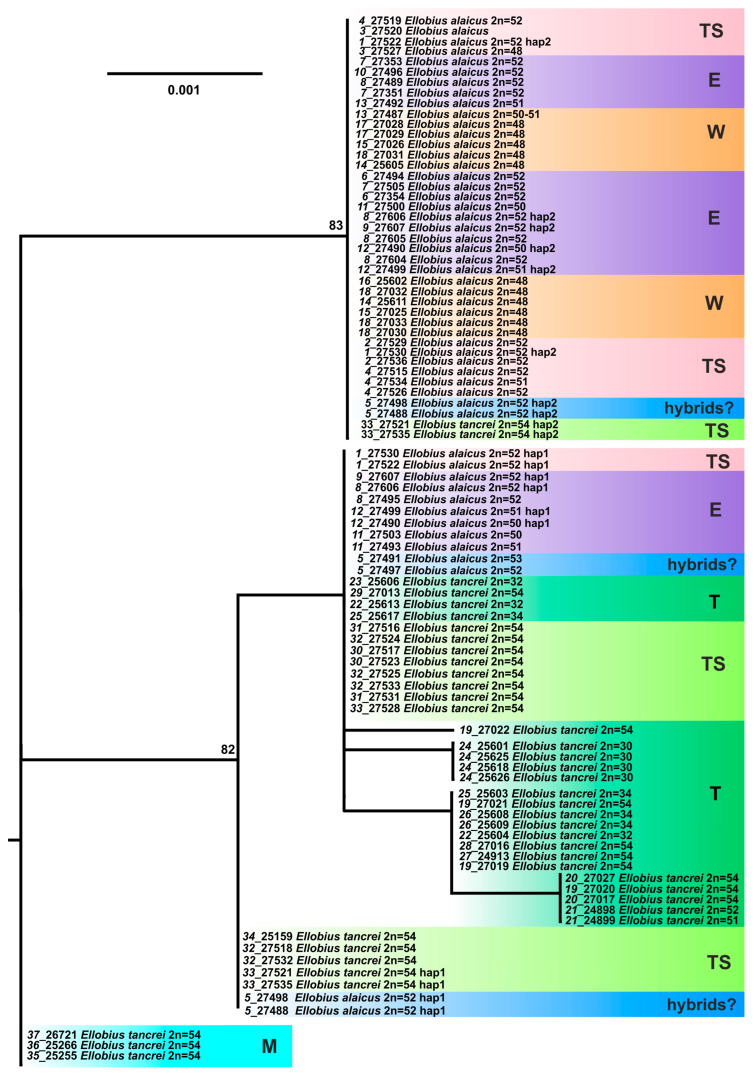
The maximum-likelihood tree based on the *IRBP* gene sequences of 82 specimens of *E. tancrei* and *E. alaicus*. Samples from *E. tancrei* Mongolian populations were used as an outgroup. Numbers above the nodes correspond to bootstrap support; values <70 are not specified. Sample names are presented as “locality number—individual number”. The color selection and abbreviations are denoted as specified in Figure 3. The haplotypes that were hypothetically derived from heterozygous genotypes are labeled as “hap 1” (haplotype, characteristic of most *E. tancrei* specimens) and “hap 2” (haplotype, characteristic of most individuals of *E. alaicus*).

**Figure 8 life-13-01751-f008:**
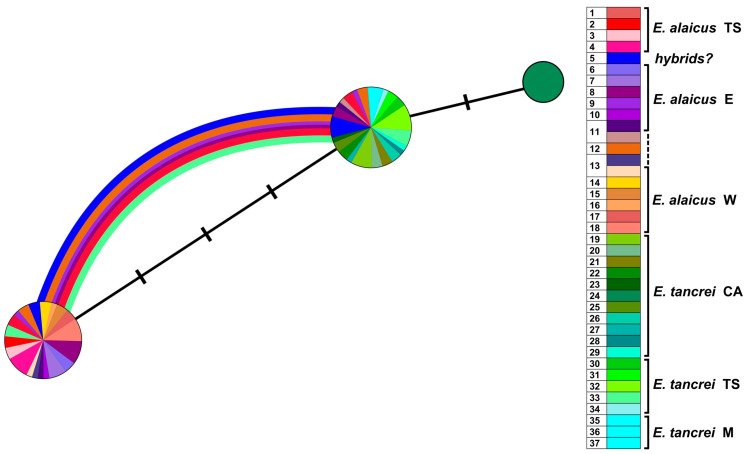
Haplotype network constructed by HaplowebMaker based on the *IRBP* gene sequences of 82 specimens of *E. tancrei* and *E. alaicus* (see Appendix A). A sample from each locality is colored with its own color; close shades unite species/intraspecific forms, according to investigated taxonomic hypotheses. The circles represent haplotypes; the curves between them indicate hybrid individuals, carrying both connected haplotypes. The area of the circles and the thickness of the curves are proportional to the number of individuals carrying the haplotype; nucleotide substitutions are indicated by dashes.

## Data Availability

GenBank accession numbers were as follows: OR231548–OR231573 for *cytb* gene, OR232598–OR232679 for the *COI* gene, OR231574–OR231599 for the *IRBP* gene, OR231600–OR231625 for the *XIST* gene first fragment, and OR231626–OR231651 for the *XIST* gene second fragment.

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
