# Peer review of "Speciation on the Roof of the World: Parallel Fast Evolution of Cryptic Mole Vole Species in the Pamir-Alay—Tien Shan Region"

_life, 2023, doi:10.3390/life13081751_

Round 1
Reviewer 1 Report
Within the framework of the study, the authors try to understand the mechanisms of rapid speciation in cryptic species of mole voles. This is a very interesting task, and a well-chosen object for its solution. Authors use a wide range of techniques - analysis of two mitochondrial and two nuclear genes, cytogenetic methods. The results seem very interesting. In the future, it would be useful to involve an ecological niche modeling (which the authors themselves indicated in the conclusion). The idea of habitat fragmentation, although it looks quite logical, could be beautifully confirmed in such a way. In addition, hypotheses based on climatic changes are put forward in the discussion, which so far look quite unfounded. As is seems to me, the analysis of fast evolving microsatellite loci also should be useful.
I have no serious negative comments to the study. The only thing I would like to mention - Discussion is very extensive and difficult to understand. It seems that structuring this section will be very appropriate, and will make it more logical and easy to read.
In addition, I have a number of minor edits.
line 3 – there is no need for a dot at the end of the title
line 12 – it is better to use chromosomal variation instead of variability
lines 46, 51, 387, etc – it is better not to start the sentence with the abbreviation of genus name. Author and year of taxa description should be added for the first mention of species name
line 48, etc. – 2n, n should be in italic
line 55 – why Kya (not kya)?
line 61, etc. – probably it is better to use «geographic range» instead of just «range»
line 102 – cytochrome (not cytochrom), and it is better to spell gene names uniformly throughout the manuscript (in upper/lower case for each gene), see also lineages 497-500 (and add the italic)
line 106 – cytochrome c oxidase subunit 1, spell it uniformly (see line 501 additionally).
lines 120-122 – have you already submit sequences to the GenBank?
lines 135-137 – “Before statistical analysis, the non-extended deletions were filled in by random nucleotides, which were absent in the total mole vole sample; this solution has been applied in our previous publications. I didn't understand this phrase. Why you used random nucleotides instead of "N"?
line 153 – did you try to use partitioning by codon position?
As a suggestion, if possible, it is better to make a karyogram more contrast, in this version bands are difficult to distinguish.
At the haplotype network for mt genes, the number of substitutions between haplotypes is not visible. It seems that the network is not scaled. It would be easier to perceive the figure if the lengths of the branches were scaled (or with digital notes on the number of substitutions)
Try to add Bayesian posterior probabilities to ML phylogenies in the main text of the article (where the topology does not differ from ML one), so as not to force the reader to open the supplementary once again
lines 232, 333 - It seems to me that when describing a tree, the phrase “star-like structure” is inappropriate even in quotation marks. Maybe polytomy?
lines 233-237. Split the sentence into two ones, if possible. The next paragraph (238-244) is also very difficult to understand. Try to break it down into shorter sentences.
line 408 – why did not you place the multilocus network in the main text? If it is so important, that it was placed in the graphic abstract
Author Response
The authors are grateful to the reviewers whose comments were very useful for our manuscript improvement. We have carefully considered the comments while preparing the revised manuscript.
Review 1.
Within the framework of the study, the authors try to understand the mechanisms of rapid speciation in cryptic species of mole voles. This is a very interesting task, and a well-chosen object for its solution. Authors use a wide range of techniques - analysis of two mitochondrial and two nuclear genes, cytogenetic methods. The results seem very interesting. In the future, it would be useful to involve an ecological niche modeling (which the authors themselves indicated in the conclusion). The idea of habitat fragmentation, although it looks quite logical, could be beautifully confirmed in such a way. In addition, hypotheses based on climatic changes are put forward in the discussion, which so far look quite unfounded. As is seems to me, the analysis of fast evolving microsatellite loci also should be useful.
Thank you for the ideas for future studies. Indeed, we plan to continue the work, involving into the analyses both a new material from northeastern Kyrgyzstan and other molecular markers (including microsatellites, above all, for study of hybridization of E. alaicus and E. tancrei as well for isolated mole vole population revealing). Moreover, ecological niche modeling is considered by us as one of our future work tasks.
I have no serious negative comments to the study. The only thing I would like to mention - Discussion is very extensive and difficult to understand. It seems that structuring this section will be very appropriate, and will make it more logical and easy to read.
We made some style corrections and changes to the arrangement of text fragments. One of paragraphs in the Discussion chapter was shifted to more fit place. We hope that the Discussion topic is now clearer.
In addition, I have a number of minor edits.
line 3 – there is no need for a dot at the end of the title
Corrected.
line 12 – it is better to use chromosomal variation instead of variability
Commonly, the total spectrum of some genetic mutations (chromosomal, nucleotide etc) in a species is determined by the “variability” term. “Variation” term designates further concrete mutations, so we prefer to keep “variability” in the case.
lines 46, 51, 387, etc – it is better not to start the sentence with the abbreviation of genus name. Author and year of taxa description should be added for the first mention of species name
Corrected.
line 48, etc. – 2n, n should be in italic
Corrected.
line 55 – why Kya (not kya)?
Corrected.
line 61, etc. – probably it is better to use «geographic range» instead of just «range»
In the context, the commonly used term “range”, as an area of distribution of a species, proposes just a geographic aspect, so, in our opinion, it is not necessary to be underlined.
line 102 – cytochrome (not cytochrom), and it is better to spell gene names uniformly throughout the manuscript (in upper/lower case for each gene), see also lineages 497-500 (and add the italic)
line 106 – cytochrome c oxidase subunit 1, spell it uniformly (see line 501 additionally).
Thank you very much for the comment. All the typos were corrected.
lines 120-122 – have you already submit sequences to the GenBank?
Yes, all newly obtained sequences (which were not previously published) were submitted to GenBank; their deposit numbers were included to the corrected versions of the manuscript and Table S1.
lines 135-137 – “Before statistical analysis, the non-extended deletions were filled in by random nucleotides, which were absent in the total mole vole sample; this solution has been applied in our previous publications. I didn't understand this phrase. Why you used random nucleotides instead of "N"?
“N” is commonly used for designation of ill-defined or non-defined nucleotides, deletions (i.e. gaps in sequences, non-coding proteins as a rule; the XIST gene is just such case) are marked in by dash. In both cases, such sites cause problems for statistical treatment because the majority of programs may not interpret “N” and “-“ symbols and delete these sites from the total sample. Deletions are just as important changes in genome evolution as nucleotide substitutions, so they should be counted. The problem is discussed in literature, but only several algorithms were elaborated to implicate indels into analysis (Awasthi, S., Mahadani, A. K., Sanyal, G., & Bhattacharjee, P. (2020). International Conference on Computation, Automation and Knowledge Management (ICCAKM) Amity University; Mahadani, A. K., Awasthi, S., Sanyal, G., Bhattacharjee, P., Pippal, S. (2021). Indel-K2P: a modified Kimura 2 Parameters (K2P) model to incorporate insertion and deletion (Indel) information in phylogenetic analysis. Cyber-Physical Systems, 1 (13), P. 2333-5777.). Among programs, which we know and use, only Mega provides a tool, which allows optionally to involve deletions to analysis or to ignore them, but even the tool appeared to be not very dependable and therefore needs a thorough use. IQTree, which was used by us in the present work, seems to ignore deletions and insertions because calculations done in two different ways (i.e. with gaps in sequences and random nucleotides instead them) demonstrated visibly more short branches, which correspond to sequences with gaps. So, we filled in deletions by random nucleotides to be sure that deletions will be involved to distance calculations and tree buildings. In our opinion, the approach that we applied here allows a correct estimation of genome changes, including indels.
line 153 – did you try to use partitioning by codon position?
For model determination and further statistical treatment we used all nucleotide substitutions, in all three codon positions simultaneously. It seems to be optimal for closely related species and intraspecific forms because it significantly increases “a phylogenetic signal”.
As a suggestion, if possible, it is better to make a karyogram more contrast, in this version bands are difficult to distinguish.
Indeed, the karyograms came out dark due to conversion the graphical files to pdf format. Corrected.
At the haplotype network for mt genes, the number of substitutions between haplotypes is not visible. It seems that the network is not scaled. It would be easier to perceive the figure if the lengths of the branches were scaled (or with digital notes on the number of substitutions)
Corrected. The dashes designating the number of substitutions were made more visible.
Try to add Bayesian posterior probabilities to ML phylogenies in the main text of the article (where the topology does not differ from ML one), so as not to force the reader to open the supplementary once again
Unfortunately, it is quite difficult to be performed because of a lack of places near many nodes. Moreover, ML and Bayesian trees slightly differ in set of some clusters, for example, Mongolian E. tancrei.
lines 232, 333 - It seems to me that when describing a tree, the phrase “star-like structure” is inappropriate even in quotation marks. Maybe polytomy?
Thank you for the comment. Corrected.
lines 233-237. Split the sentence into two ones, if possible. The next paragraph (238-244) is also very difficult to understand. Try to break it down into shorter sentences.
Corrected.
line 408 – why did not you place the multilocus network in the main text? If it is so important, that it was placed in the graphic abstract
Unfortunately, the manuscript size and the number of figures, which were included in the main text, are already quite large. So, trees and nets, which display the variability most clearly and entirely, were included by us in the main text. Multilocus nets smooth the peculiarities of different gene polymorphisms, so we preferred to shift them into supplementary materials. Otherwise, multilocus net reflects (although not quite obviously) all most interesting results obtained by us (significant differentiation within both mole vole species, hybridization of them and intraspecific forms of E. alaicus) and may serve a visual illustration, summarizing of main results and so fit for graphical abstract.
Reviewer 2 Report
Dear authors,
Please find attached my review of your manuscript. I have two main concerns:
1) please revise the length of COI sequence.
2) Whenever suggesting cryptic speciation make it sure that from the molecular analyses performed, only Bayesian and Maximum-likelihood of cytb and Bayesian of COI sustain it.
Moreover, many corrections and suggestions were directly made in the pdf file. I strongly suggest you to deal with all of them for a better presentation of your important work.
Regards

Few sentences should be checked, all marked in the pdf file.
Author Response
The authors are grateful to the reviewers whose comments were very useful for our manuscript improvement. We have carefully considered the comments while preparing the revised manuscript.
Review 2.
Dear authors,
Please find attached my review of your manuscript. I have two main concerns:
1) please revise the length of COI sequence.
The first half of the cytochrome oxidase gene (about 700 bp) is commonly used in the species genetic passportization ('barcode') scientific program. However, the entire gene is significantly longer. In voles and mole voles its total length is just over 1540 bp (see, for example, GenBank records KP190312 and NC_054160).
2) Whenever suggesting cryptic speciation make it sure that from the molecular analyses performed, only Bayesian and Maximum-likelihood of cytb and Bayesian of COI sustain it.
Although taxonomical interpretation of our latest results is really ambiguous, we still stick to the point of view, that E. alaicus and E. tancrei are very young species as they have different pools of gene and chromosome variations. Moreover, hybridization of them seems to be restricted because we know only several reliably confirmed examples of it. Nuclear gene evolution goes less rapidly compared with mitochondrial genes, so a lack of fixed differences in the IRBP and XIST sequences of E. alaicus and E. tancrei may not be the final proof of their conspecificity. In addition, we don’t know yet how different other nuclear genes are. Resolving taxonomical problems in the case needs additional studies.
Moreover, many corrections and suggestions were directly made in the pdf file. I strongly suggest you to deal with all of them for a better presentation of your important work.
Regards
We got carefully acquainted with the comments and we would like to express our gratitude to the reviewer for them. For the majority of the comments, we made corrections in the manuscript.
line 11 – “were described” was replaced by “were revealed”;
line 19 – corrected;
lines 21, 67 – in these cases, phyla or phylogenetic lines seem to be more optimal because they don't need to address to cladistical analysis results;
line 42 – the information that the Ellobius genus corresponds to fossorial cricetid rodents was added;
line 44 – reference to Lebedev et al. (2020) was added;
line 57 – “within one generation of researchers” was concretized as “within approximately 30 years, i.e. one generation of researchers”;
line 87 – “new animals” was replaced by “newly collected animals”;
line 127 – preliminary species identification was performed on the basis of the karyotyping. The explanation was added to the "Sampling" chapter;
line 128 – text in this line was replaced by the following one: “for the statistical analyses of variability of the cytb, COI, IRBP, and XIST genes as well for building nets and phylogenetic trees”;
line 135-136 – theoretically, filling out deletions by random nucleotides may visibly change branch lengths and even tree topology (that, however, did not happen in our case). Otherwise, in our opinion, significantly more bias arises if deletions, which are serious genome changes, are not included in an analysis. The majority of programs used for phylogenetic analysis cannot treat symbol "-", designating deletion, and exclude sites with deletion/insertion from the total sample. We filled in deletions by random nucleotides to be sure that deletions will be maintained in statistical analysis;
line 152 and further throughout the text – “dendrogram” was replaced by “tree”;
lines 209, 253, 254 – unfortunately, the manuscript size is already quite large, so we had to include into the text the most important of figures. For the same reason, some of the most important genetic distance values were presented in the text; the rest were maintained in tables in supplementary materials;
lines 210-213 – the sentence was rewritten as following: “The analysis of mitochondrial cytb gene in the total sample of mole voles (including previously studied ones) demonstrated high differentiation within E. tancrei and E. alaicus, at that samples from Tien Shan, representing divergent intraspecific genetic forms, made significant contribution to genetic variability of both species”;
line 214 – corrected;
line 217 – information about karyotypes of E. tancrei specimens from Tajikistan was added;
lines 224-225 – the final part of the sentence was changed by following: “are comparable with differences between E. alaicus and E. tancrei from Central Asia (0.026)”;
line 229 – polytomy in ML-tree built on the COI analysis data is discussed below;
line 232 and further throughout the text – “star-like structure” was replaced by “polytomy”;
Lines 233- 244 the text was rewritten as
"The analysis of the concatenated sequences of both mitochondrial genes levelled the inconsistency and produced Bayesian and ML trees with similar topologies (Figures 3, S6), like those in the cytb gene study (Figures S2, S3). Despite this, it should be concluded that a clusterization pattern for the intraspecific forms of E. alaicus and E. tancrei may significantly vary, depending on different mitochondrial markers".
"Likewise the cytb gene, the average genetic distances, which were calculated using the COI gene fragment and joined sequence of two mitochondrial markers, between E. alaicus and E. tancrei from Central Asia (0.020 for the COI gene and 0.023 for concatenated mitochondrial sequence, see Table S2), insignificantly excess differences determined between major subclades within E. alaicus (maximally 0.017 for the COI gene and 0.020 for concatenated mitochondrial sequence) and E. tancrei from Central Asia (maximally 0.015 and 0.019 respectively)"
lines 258-259 – the sentence was rewritten as 'haploweb based on cytb and COI sequences allowed to demonstrate numerous mutational steps between intraspecific forms and species along with high level of their polymorphism'.
line 256 – “haplotypic diversity” that is the specific statistical parameter was replaced by “haplotypic polymorphism”. Thank you for the comment;
line 321 – as we know, determination of the basal node support is practically impossible because of absence of a branch or clade on which such calculation may be performed. Polytomy in the case of the IRBP gene is discussed below;
lines 381-382 – corrected;
lines 409-413 – the sentences were corrected;
lines 422-423 – of course, we share your opinion, that Mongolian populations deserve to be described as a separate species and the materials from type locality (vicinities of Zaisan Lake) is necessary for it. Our colleagues just obtained samples from there;
line 436 – “specialized” was replaced by “divergent”;
line 438 – “Specialization” was replaced by “Independent mutation accumulation”;
line 536 – “active radiation” was replaced by “significant differentiation”.
lines 555-6 this is consistent only for Bayesian and ML trees of cytb and Bayesian tree of COI. If you want to sustain that position of cryptic species, it should by clear that you have evidence only for these trees. – We have strengthened the focus on this point because we believe that chromosomal rearrangements play the leading role in the divergence of this group. References to publications emphasis the role of such rearrangements due to changes in linkage groups and accelerated evolution of the genetic content of the altered chromosomes and all the genome. As these species are young, the first signs of DNA alterations were evident at the Bayesian and ML trees of cytb and Bayesian tree of COI.

Round 2
Reviewer 2 Report
Dear authors, just check the indication of substititutions of "dendrograms" by "trees" in pages 16 and 17, specially in the proof. Congratulations for the very nice work, I hope to see it published soon!
Best wishes